# A Flow Velocity Measurement Method Based on a PVDF Piezoelectric Sensor

**DOI:** 10.3390/s19071657

**Published:** 2019-04-06

**Authors:** Qi Li, Junhua Xing, Dajing Shang, Yilin Wang

**Affiliations:** 1Acoustic Science and Technology Laboratory, Harbin Engineering University, Harbin 150001, China; leechi319@163.com (Q.L.); xjh0315@163.com (J.X.); yiyilinlin@outlook.com (Y.W.); 2Key Laboratory of Marine Information Acquisition and Security (Harbin Engineering University), Ministry of Industry and Information Technology, Harbin 150001, China; 3College of Underwater Acoustic Engineering, Harbin Engineering University, Harbin 150001, China

**Keywords:** velocity measurement, PVDF film

## Abstract

To measure the flow velocity of a fluid without affecting its motion state, a method was proposed based on a polyvinylidene fluoride (PVDF) piezoelectric film sensor. A self-made PVDF piezoelectric sensor placed parallel with the flow direction was used to measure the flow velocity. First, the piezoelectric characteristics of PVDF were obtained theoretically. Next, the relationship between flow velocity and sound pressure was verified numerically. Finally, the relationship between flow velocity and the electrical output of the PVDF piezoelectric film was obtained experimentally. In conclusion, the proposed method was shown to be reliable and effective.

## 1. Introduction

The determination of fluid flow velocity is not only the basis for researchers to analyze fluid motion characteristics and motion laws but also the key to studying the spatial distribution of the flow field and how the components of the flow field interact. The flow rate is measured mainly with a current meter. The first-generation instrument, namely the rotary flow meter, is widely applicable and relatively inexpensive. The technology has matured with continuous improvement [1,2], but such instruments must be inserted into the fluid to measure the flow field, thereby causing interference. The second-generation ultrasonic Doppler velocimeter is a non-contact, high-precision instrument that has been studied widely as a promising instrument [3,4,5], but it is relatively expensive and cannot be used in environments with large sediment concentration.

In 1969, Kawai discovered that polarized polyvinylidene fluoride (PVDF) has piezoelectric properties [6], a discovery that attracted a great deal of attention from industry. After comparing different piezoelectric materials, Kowbel et al. found that the piezoelectric coefficient *g*_31_ of PVDF is much higher than those of other piezoelectric materials [7], and confirmed its excellent sensing performance. The low cost and high performance of PVDF materials make them popular in various engineering and manufacturing fields. van Tol and Hughes made an underwater sound-intensity probe using PVDF material [8]. Lu et al. made PVDF sensors and used them to detect surface acoustic waves [9]. By measuring its linearity, Yuan et al. demonstrated that PVDF piezoelectric film is highly suitable for measuring dynamic pressure [10]. Ge et al. used PVDF film for the first time to measure turbulent pressure pulsations, and they derived a general expression for the self-power spectral density of the output voltage of a planar PVDF hydrophone in a turbulent boundary layer (BL) with pressure fluctuations [11]. Wang et al. developed a PVDF piezoelectric array of pressure sensors and studied the impulsive pressure generated by cavitation bubble collapse [12]. Yang et al. used flapping wings with PVDF sensors to detect the lifting force and to modify the aerodynamic forces of a micro aerial vehicle [13]. Tanaka et al. developed a wearable tactile sensor for measuring skin vibrations using a PVDF film [14]. Dung and Sasaki used numerical simulations to obtain the output response of a PVDF sensor attached to a cantilever beam subjected to impact loading [15]. Li used PVDF sensors to measure flow velocity, but the sensor surface was perpendicular to the flow direction, creating an unavoidable effect on the fluid motion [16].

The theory of turbulent BLs is an important branch of theoretical fluid mechanics. Turbulence is generated when air flows past the surface of a plate, and fluctuating pressure arises because of the interaction between the turbulence and the surface. Kline et al. described the structure of turbulent BLs and showed that turbulent BLs on smooth walls can have zero, negative, and positive pressure gradients [17]. Abu-Ghannam and Shaw investigated the natural transition of a BL on a flat plate in a free stream in a low-speed wind tunnel, and proposed empirical relationships for predicting the start and end of the transition [18]. Willmarth measured the fluctuating pressure of a turbulent BL using a barium titanate microphone and demonstrated relevant properties about the turbulent BL [19]. Ling solved some problems regarding the fluctuating pressure in the turbulent BL on a submarine model by means of finite-element numerical simulation [20]. Shang et al. studied the fluctuating pressure on the surface of an underwater airfoil structure and established that there was a relationship between the velocity and the magnitude of the pressure pulsation [21]. Abshagen et al. performed underwater experiments on the flow-induced noise in the interior of a towed body generated from the surrounding turbulent BL and analyzed the scaling relations of the wall pressure fluctuations, interior hydroacoustic noise, and a reduction of pressure fluctuations through the plate [22]. According to the principles of fluid mechanics, the magnitude of the fluctuating pressure changes with the fluid velocity, and the pressure fluctuation at different flow rates can be measured by a PVDF film. The flow velocity can be measured by specifying the relationship between the output of the PVDF film and the fluid velocity. The theory can then be extended from the case of air to the case of water. Because the PVDF film is flat and the fluid flows past the surface, there is less impact on the fluid motion. The present paper studies how to measure different air speeds using PVDF piezoelectric film sensors.

## 2. Theory

### 2.1. Piezoelectric Principle of PVDF Film

PVDF is a semi-crystalline polymer, the β-type crystalline form being the most polarized [23]. It has excellent piezoelectric, thermoelectric, and ferroelectric properties [24] and can be used to make PVDF sensors. According to the piezoelectric principle, the first type of piezoelectric equation in the absence of an applied electric field is:(1){D}=[d]{T},
where {*D*} is the electrical displacement, {*T*} is the stress, and the matrix of piezoelectric constants of the PVDF film is:(2)$$\left[d\right]=\left[\begin{array}{cccccc} 0 & 0 & 0 & 0 & 0 & 0\\ 0 & 0 & 0 & 0 & 0 & 0\\ d_{31} & 0 & d_{33} & 0 & 0 & 0 \end{array}\right].$$

Because only three directions of the film are subjected to stress, the parameter *T*_1_ is zero. Combining Equations (1) and (2) gives: (3)$$D_{3}=d_{33}T_{3}.$$

In the present experiments, the entire outer surface of the PVDF piezoelectric film senses force, the area being *A*. Thus, we have:(4)$$Q=D_{3}A=d_{33}T_{3}A.$$

Therefore, the charge *Q* generated on the surface of the PVDF film is proportional to the external force applied in the *Z* direction (see Figure 1) when *A* is a constant parameter.

When fabricating a PVDF piezoelectric sensor, electrodes are plated on the upper and lower surfaces of the film as shown in Figure 2. Charges of different polarities generated by the film accumulate spontaneously on the electrodes on the upper and lower surfaces to form the most basic rectangular capacitive structure. At this time, the film can be approximated as a parallel-plate capacitor model. The distance between the two plates is *δ* (m), the area of the plate is *A* (m^2^), and the amount of accumulated charge on the upper and lower plates is *q* (C).

The surface charge density of a unipolar plate is:(5)$$\sigma =\frac{q}{A},$$
and the electric field strength is:(6)$$E=\frac{\sigma }{\varepsilon_{r}\varepsilon_{0}},$$
where $\varepsilon_{r}$ is the relative dielectric constant of the PVDF film and ε_0_ ≈ 8.85 × 10^−12^ F/m is the vacuum dielectric constant. The potential difference between the two poles of the film is: (7)$$\Phi_{AB}=\int_{A}^{B}Edl=E\delta =\frac{\sigma \delta }{\varepsilon_{r}\varepsilon_{0}}=\frac{q\delta }{\varepsilon_{r}\varepsilon_{0}A},$$
and the charge capacity is:(8)$$C_{AB}=\frac{q}{\Phi_{AB}}=\frac{q}{\left(\frac{q\delta }{\varepsilon_{r}\varepsilon_{0}S}\right)}=\frac{\varepsilon_{r}\varepsilon_{0}A}{\delta }.$$

The voltage between the two poles of the film is:(9)$$U=\frac{Q}{C_{AB}}.$$

Combining Equations (4), (8), and (9) gives:(10)$$U=\frac{d_{33}\delta }{\varepsilon_{0}\varepsilon_{r}}T_{3},$$
and the film sensitivity is:(11)$$M=\frac{U}{T_{3}}=\frac{d_{33}\delta }{\varepsilon_{0}\varepsilon_{r}}.$$

The thickness of the PVDF piezoelectric film is known, thus the sensitivity is only a certain constant and no independent measurement is required. Therefore, the output voltage between the two poles of the PVDF piezoelectric film is directly proportional to the external stress received in the third direction, and the stress on the surface of the film can be obtained by measuring that electrical output.

### 2.2. Acoustic Principle of Turbulent Fluctuating Pressure

The PVDF film can be approximated as a smooth thin plate that satisfies the no-slip condition. When a viscous fluid flows past the surface of the smooth thin plate, a thin shear layer known as a BL forms near the surface, and different BL regions can be divided according to different flow states, namely a laminar flow zone, a transition zone, and a turbulent zone.

Fluid motion produces hydrodynamic noise. For a smooth thin plate, there are two types of flow noise. The tiny vortices that form and detach continuously in the turbulent flow form a noise source (a quadruple sound source), and the noise generated by it radiates directly to the flat plate. The acoustic–solid coupling effect causes the flat structure to vibrate, and that noise excitation is called acoustic wall pressure fluctuation (AWPF). The turbulence acting on the flat plate generates noise directly by vibrating the flat structure through fluid–solid coupling, and that noise excitation is called turbulent wall pressure fluctuation (TWPF). The two types of flow noise coexist and are indispensable for exciting vibration of the PVDF film. In general, flow noise is one of three main sources of noise for underwater vehicles. Its intensity increases rapidly as the vehicle speed increases, with the radiated sound power being proportional to the fifth to seventh power of the speed [25].

We propose using simulation to verify the correctness of turbulent BL theory and to determine the relationship between the plate flow-excitation noise intensity and the flow velocity.

### 2.3. Experimental Principle

Sound pressure is the change after the atmospheric pressure is disturbed by a sound wave and is defined as *P_e_* = *P*−*P*_0_. In this formula, *P* is the air pressure after disturbance by the sound wave and *P*_0_ is the initial air pressure. When the flow field becomes turbulent and produces acoustic radiation, the air pressure above the film changes continuously. According to the piezoelectric effect of the PVDF film, the stress on the PVDF film is the amount of pressure change, that is, the sound pressure.

The sound pressure level (*SPL*) is given by: (12)$$SPL=20\times \log\frac{P_{e}}{P_{ref}}(\mathrm{dB}),$$
and the sound intensity level *SIL* is given by:(13)$$SIL=10\times \log\frac{I}{I_{ref}}(\mathrm{dB}).$$

The relationship between sound intensity and sound pressure is expressed as:(14)$$I=\frac{P_{e}}{\rho_{0}c_{0}}\left({W/m}^{2}\right).$$

Therefore, the relationship between sound intensity level and sound pressure level is expressed as:(15)$$SIL=10\times \log\frac{I}{I_{ref}}=10\times \log(\frac{P_{e}{}^{2}}{\rho_{0}c_{0}}\times \frac{400}{P_{ref}{}^{2}})=SPL+10\times \log\frac{400}{\rho_{0}c_{0}},$$
where *P_e_* is the effective value of the sound pressure to be tested, the value of *P_ref_* is 2 × 10^−5^ Pa, *I* is the sound intensity to be tested, and the value of *I_ref_* is 10^−12^ W/m^2^. Under normal circumstances, the sound intensity level and the sound pressure level differ by only a small correction value of log(400/(*ρ*_0_*c*_0_)). The present experiments were conducted in air at room temperature, thus the difference is negligible:(16)$$SIL=10\times \log\frac{P_{e}{}^{2}}{P_{ref}{}^{2}}=10\times \log\frac{{\left(\frac{U}{M}\right)}^{2}}{P_{ref}{}^{2}}=10\times \log U^{2}-10\times \log M^{2}-10\times \log P_{ref}{}^{2}.$$

From Section 2.1, we have that $U=\frac{d_{33}\delta }{\varepsilon_{0}\varepsilon_{r}}T_{3}$ and $M=\frac{U}{T_{3}}=\frac{d_{33}\delta }{\varepsilon_{0}\varepsilon_{r}}$. Therefore, the difference between the sound intensity and 10 log*U*^2^ is only a fixed constant, and the sound intensity levels obtained in the experiments and the simulation should have the same power relationship with the flow velocity *v*.

## 3. Finite-Element Simulation of Flow Excitation Noise

### 3.1. Flow-Field Modeling and Meshing

To provide a stable wind field, a cylindrical tubular fluid domain with a radius of 0.08 m and a length of 0.74 m was created using the software ICEM, as shown in Figure 3, and was used as the region through which the air flowed. The PVDF film was placed in the middle of a polyvinyl butyral plastic plate with a length of 0.54 m, a width of 0.108 m, and a thickness of 0.15 m. The PVDF film itself was 0.14 m long, 0.108 m wide, and 0.01 m thick. Table 1 lists the various boundary conditions. An unstructured grid was used throughout the entire domain and local grid refinement was done in the BL region near the slab. By controlling the height of the near-wall grid, the range of Y+ satisfied the requirements of the corresponding wall function; this model was used to simulate the parallel flow of air at different speeds.

### 3.2. Numerical Simulation of Flow Field

The finite-element analysis software FLUENT was used to simulate the flow field. Because time-domain information was required, the flow motion was considered as being unsteady. The flow was divided into two parts, namely a steady part and an unsteady part [26]. The FLUENT flow-field calculation was also divided into two parts, namely steady-state calculation and transient calculation.

Transient vortex simulation of the dynamic sub-grid model was used to calculate the transient of the flow field. Large-eddy simulation is sensitive to the initial flow-field conditions. Therefore, the steady-state calculation of the flow field was carried out first, with the converged result being used as the initial flow field for the next step. The PISO pressure-correction equation was used in the large-eddy simulation, the QUICK format was used to interpolate the flow terms, and the pressure interpolation was the PRESTO format. Once the calculation had stabilized, the flow-field numerical results were extracted, the time step being 0.0001 s, and 2000 time steps were sampled and used for the acoustic calculation.

Figure 4 shows a vortex cloud image obtained from the transient operation. The vorticity is lowest in the left-most region (near the velocity inlet, namely the laminar region), turbulence arises in the transition zone, and the vorticity becomes relatively high and stable upon entering the turbulent region. This behavior is consistent with turbulent BL theory and shows that the transient result is calculated correctly. The PVDF film is basically located in the turbulent flow zone.

### 3.3. Numerical Simulation of Sound Field

Simulating the AWPF and TWPF requires different models, and the ICEM software was used to establish them. The ACTRAN finite-element software was used to do the simulation. The results of the flow-field calculation in FLUENT and the established acoustic-field grid were both imported into the ICFD module of ACTRAN, and the time-domain analysis component [27] was added to convert the results of the flow-field calculation into time-domain information and then to calculate the result. A Fourier transform was performed after interpolation and completion of the mapping relationship [28].

The frequency response curves of the turbulent BL pressure responses at different wind speeds (10, 15, 20, 25 and 30 m/s) are plotted in Figure 5. As the wind speed is increased, the trend of the pressure response (each curve) remains basically unchanged, and the numerical values increase uniformly at different frequencies; the pressure response increases with frequency in the low-frequency range, whereas the medium-to-high frequency range decreases by 6 dB per octave as the frequency increases. Thus, the curve for each wind speed has a peak in the frequency range of 100–500 Hz. The exported ACTRAN results are given in Table 2.

According to the formula $SPL_{2}-SPL_{1}=n\times 10\times \log(\frac{v_{2}}{v_{1}})$, the average value of *n* for the different ranges of wind speed (10 to 15 m/s, 15 to 20 m/s, etc.) was *n* ≈ 6, that is, the total sound level increases linearly with the logarithm of the sixth power of the wind speed.

The above data were imported into MATLAB, and the sound pressure values at each frequency point were added to obtain the curve of the total sound pressure level against wind speed. The logarithm of the sixth power of the wind speed was used as the horizontal coordinate and the total sound level was used as the vertical coordinate. The curve is shown in Figure 6.

To clearly see the linear relationship, the horizontal coordinate in Figure 6 corresponded to the actual velocity. The simulation confirms (i) the theoretical assumption regarding sound radiation from the dipole in the turbulent BL of a flat plate and (ii) the theoretical correctness of the linear relationship between the total sound level and log (*v*^6^).

## 4. Flow-Rate Measurement Experiment and Analysis of Results

### 4.1. Production of PVDF Sensor

The PVDF sensor consisted of three parts. One was a PVDF film with a length of 0.145 m, width of 0.105 m, and height of 0.0015 m. Another part was a rectangular backing that was 0.54 m long, 0.15 m wide, and 0.01 m high. The third part consisted of two rectangular columns that were 0.54 m long, 0.03 m wide, and 0.02 m high. The PVDF film was fixed onto the rectangular column with bolts, and the column was glued to the backing, which was a hard transparent plastic plate that acted as a protective layer to prevent the deformation of the soft film from affecting the noise measurement. The PVDF sensor is shown in Figure 7.

The electrodes covering the surface of the PVDF film were divided into 4 × 3 blocks (a total of 12 array elements, each 0.028 m long and 0.02 m wide) that were insulated from one another. A diagram of the PVDF sensor array division is shown in Figure 8. The two poles of each array element required external lead wires to be connected to the measuring instrument. The film was placed on a hard transparent plastic plate that acted as a protective layer. The fabricated PVDF pressure film sensor was fixed in the round wind pipe that had a length of 3 m and an inner diameter of 1.6 m. To reduce the influence of vibrations, a thin layer of cotton wool was shock-coated and placed on the edge of the smooth thin plate for the experiment.

### 4.2. Experimental Steps

The experimental equipment was connected as shown in Figure 9 and Figure 10. The fan was connected to an air duct to provide a stable wind field, and the PVDF sensor was fixed in the air duct to avoid the sensor being displaced by the drag force, as shown in Figure 11 and Figure 12. A frequency converter was used to adjust the frequency of the fan motor to increase the wind speed from 9 to 27 m/s, with the spectra of the electrical output signals from the PVDF sensor being sampled at integer values of the wind speed using a data collector (Brüel & Kjær PULSE) and a computer. During the experiment, elements close to the rectangular columns were easily affected by the boundary, and the spectra of the electrical output signals from them were unstable. The spectra of the electrical output signals from the elements numbered 5, 6, 7, and 8 were stable and consistent, and their average was taken as an output to the sensor. Each spectrum was sampled twice and recorded as the result of Experiment 1 and Experiment 2, respectively.

### 4.3. Experimental Results

The results of the two experiments were highly consistent, and thus the results of only Experiment 1 have been described here. The spectra of the voltage output (represented by log(*V*), $\log(V)=\log(\frac{Voltages}{1\mathrm{Volt}})$, using a reference level of one volt) at different wind speeds (9, 14, 18, 21, 24 and 27 m/s) are shown in Figure 13.

As the wind speed increases, the trend of each line spectrum does not change much, and the response value increases uniformly at a different frequency. For each wind speed, the output decreases with a frequency in the mid-to-high frequency band, and the attenuation trend is almost the same. These results are consistent with the flow noise law.

Now follows is a detailed analysis of the relationship between the voltage output and flow velocity. Octave analysis was used to add the output of each frequency band, and the curve of the voltage output (represented by log(*U*), *U* is the sum of *V* at each frequency point in each frequency band) against wind speed was plotted in different frequency bands. The results are shown in Figure 14, where the horizontal coordinate is log(*v*^6^) and the vertical coordinate is log(*U*).

The relationship between log(*v*^6^) and log(*U*) was fitted in MATLAB using the least-squares method, and the fitting result satisfied the formula $10\log(U^{2})=a\cdot \log(v^{6})+b$, where *a* and *b* are constants whose values change slightly with the frequency band. For example, calculations using the best data set (the one for the 5000 Hz frequency band) gave *a* = 9.7827 and *b* = −61.4879 for 5000 Hz. Because the difference in value between sound intensity and $10\log U^{2}$ is only a fixed constant, the total sound level increased linearly with the logarithm of the sixth power of the wind speed. This behavior is consistent with the dipole-type acoustic radiation law.

The errors between the wind speed measured by the PVDF sensor (represented by *v_pvdf_*) and that measured by a standard anemometer (*v_standard_*) during the experiment can be calculated by using the formula: $Errors=\frac{v_{pvdf}-v_{standard}}{v_{standard}}\times 100\%$. The calculation results are shown in Figure 15. For each wind speed, the error was less than 5%, thus showing that the proposed method for measuring the flow rate is reliable.

## 5. Conclusions

In the present work, the piezoelectric principle of a PVDF film was analyzed, the turbulence noise of a flat-panel model was simulated, a flow velocity measurement system with a PVDF film as the sensing component was built, and the piezoelectric response of the PVDF sensor under wind excitation was measured. The results found the following:

(1) The amount of charge generated on the surface of the PVDF film and the voltage output between the two poles was proportional to the pressure in the Z direction, thereby showing the practicability of the PVDF film as a sensor for pressure measurement.

(2) The results supported the existence of a turbulent BL and the existence and distribution of pressure fluctuations. A linear relationship was calculated between the total sound level and the logarithm of the sixth power of the wind speed.

(3) The output of the PVDF sensor was measured for different wind speeds. The frequency response was analyzed, and the curve of output against wind speed was plotted for different frequency bands. The square of the voltage output (the logarithm thereof) was found to be positively correlated with the sixth power (the logarithm thereof) of the wind speed, which is consistent with the dipole acoustic radiation law. The results confirmed that the piezoelectric response law of the experimental sensor can be used for measuring flow velocity.

The measurement of fluid flow velocity without disturbing the motion state of the fluid itself was realized and has wide applications. At the same time, the research method was instructive for the study of acoustic–solid coupling and fluid–solid coupling.

## Figures and Tables

**Figure 1 sensors-19-01657-f001:**
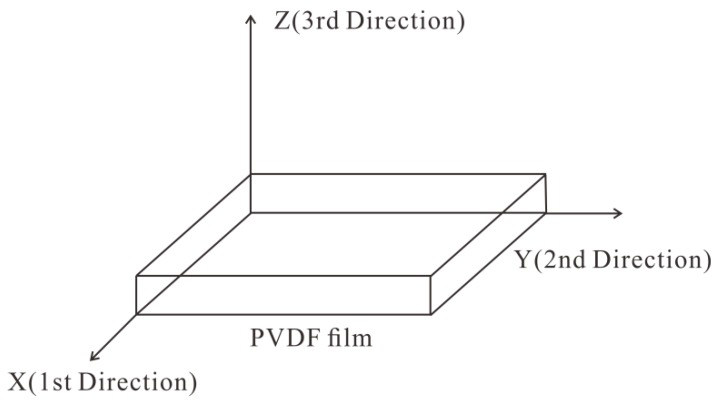
Directions relative to polyvinylidene fluoride (PVDF) film.

**Figure 2 sensors-19-01657-f002:**
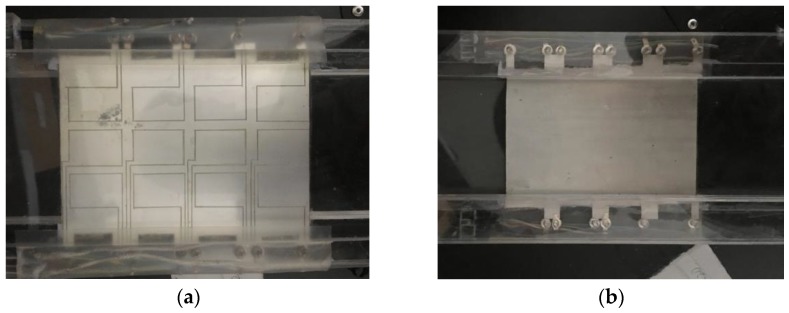
Photographs of PVDF film. (**a**) Upper surface of PVDF film, (**b**) Lower surface of PVDF film.

**Figure 3 sensors-19-01657-f003:**
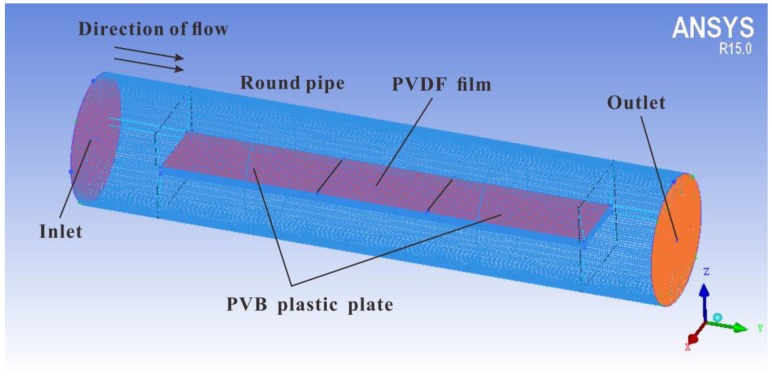
Modeling and meshing of fluid domain.

**Figure 4 sensors-19-01657-f004:**
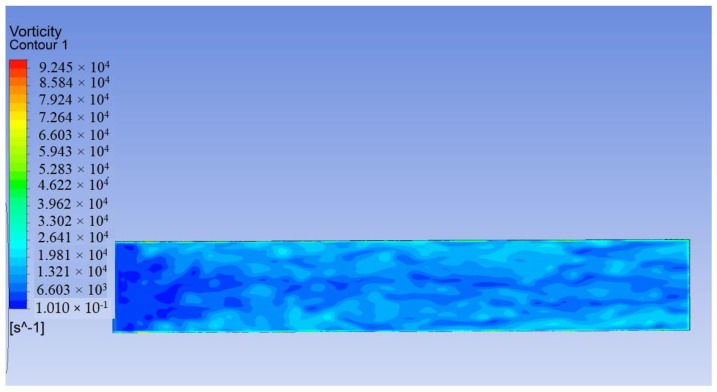
Vorticity cloud image of PVDF film.

**Figure 5 sensors-19-01657-f005:**
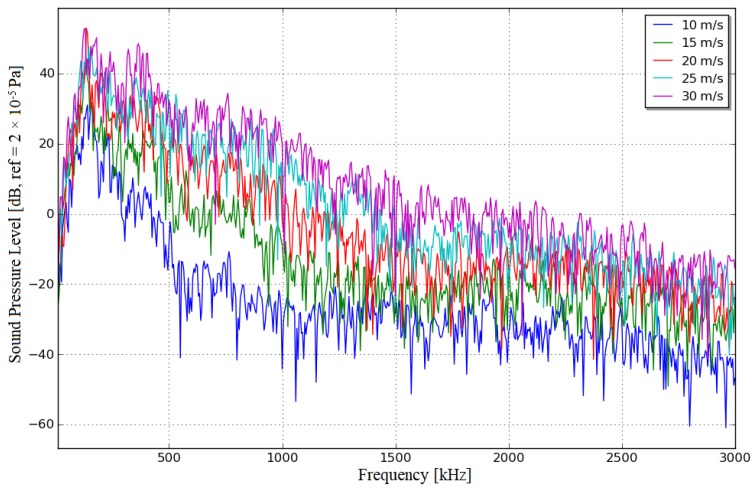
Frequency response curves for different wind speeds.

**Figure 6 sensors-19-01657-f006:**
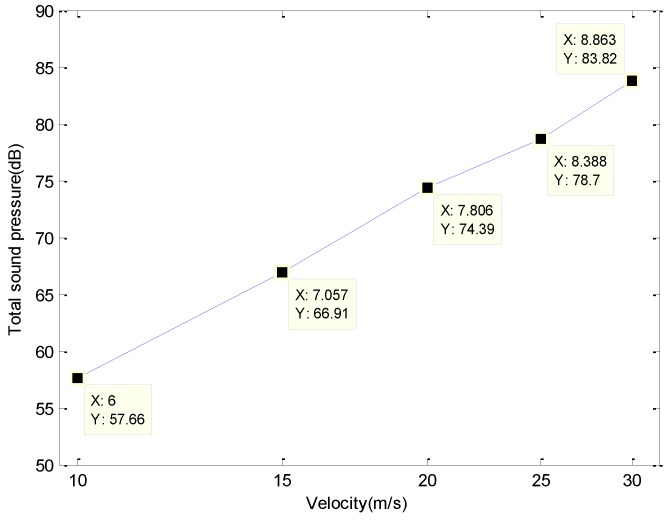
Variation of total sound level with speed.

**Figure 7 sensors-19-01657-f007:**
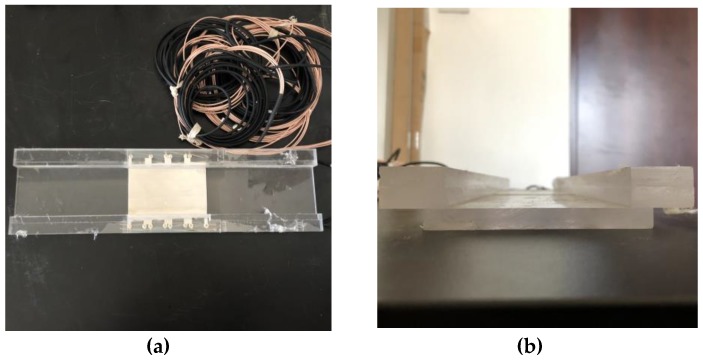
Photographs of PVDF sensor. (**a**) Top view of PVDF sensor, (**b**) Side view of PVDF sensor.

**Figure 8 sensors-19-01657-f008:**
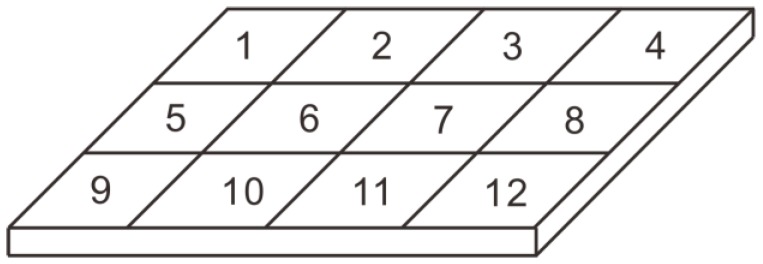
Diagram of PVDF sensor array division.

**Figure 9 sensors-19-01657-f009:**
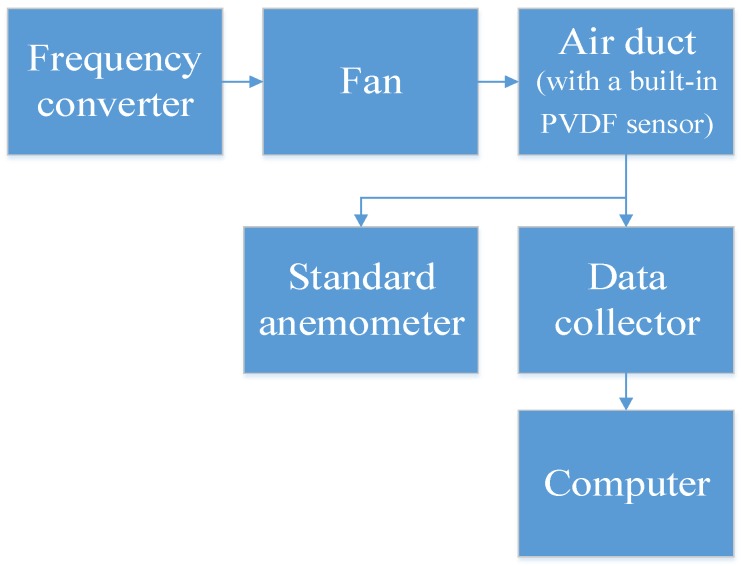
Instrument connection sequence.

**Figure 10 sensors-19-01657-f010:**
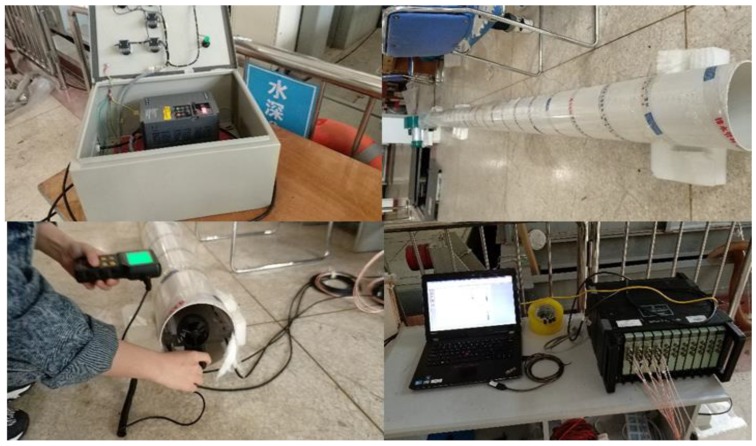
Experimental setup.

**Figure 11 sensors-19-01657-f011:**
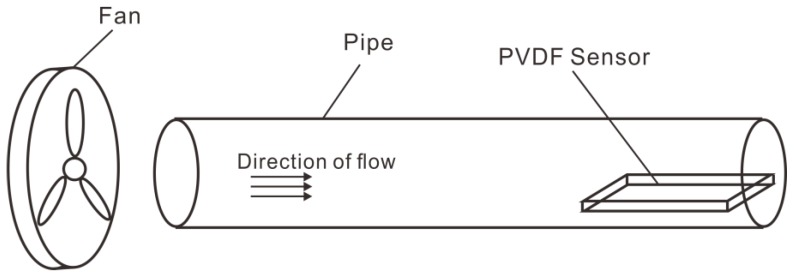
Position of PVDF sensor.

**Figure 12 sensors-19-01657-f012:**
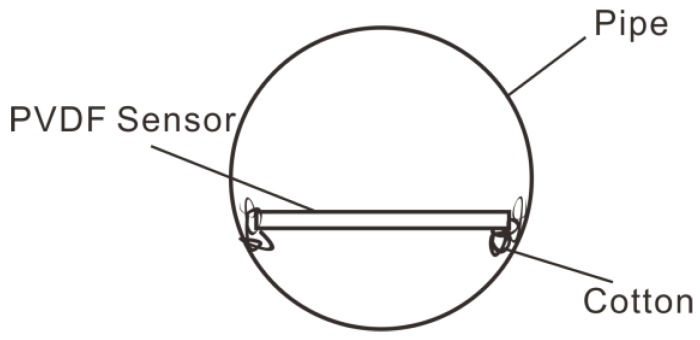
Cross section of pipe.

**Figure 13 sensors-19-01657-f013:**
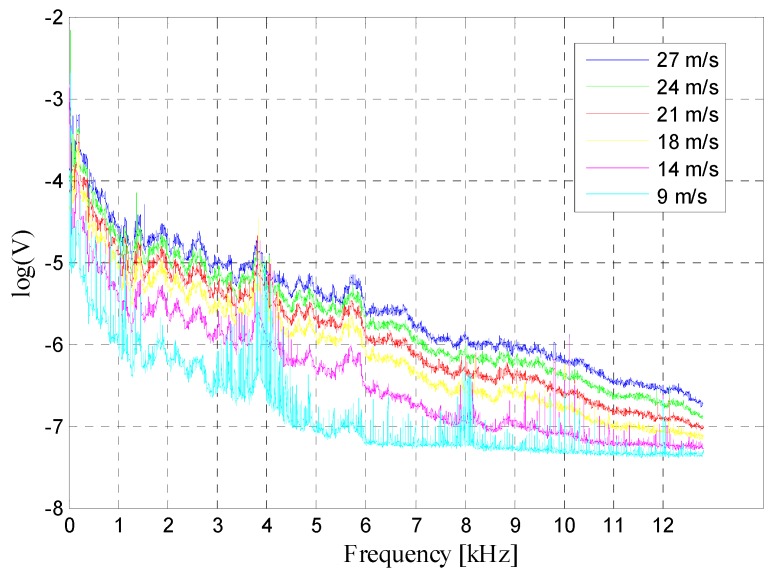
Curve of output with frequency at different wind speeds.

**Figure 14 sensors-19-01657-f014:**
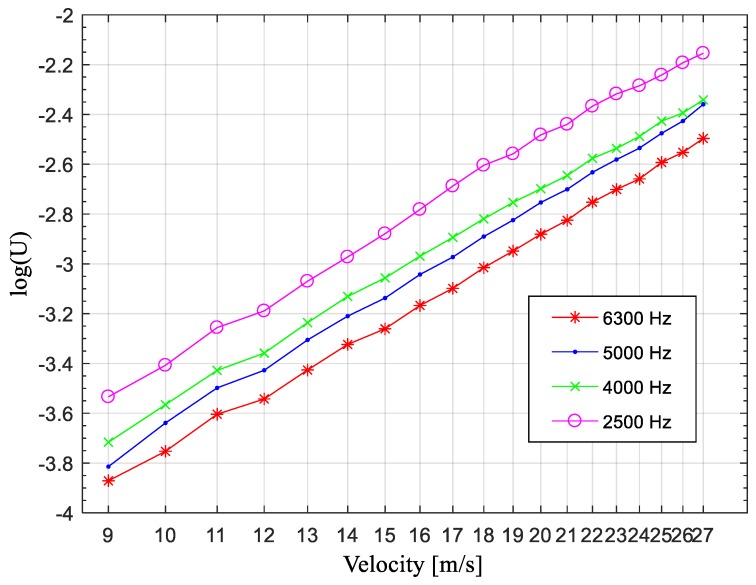
Variation of output with speed.

**Figure 15 sensors-19-01657-f015:**
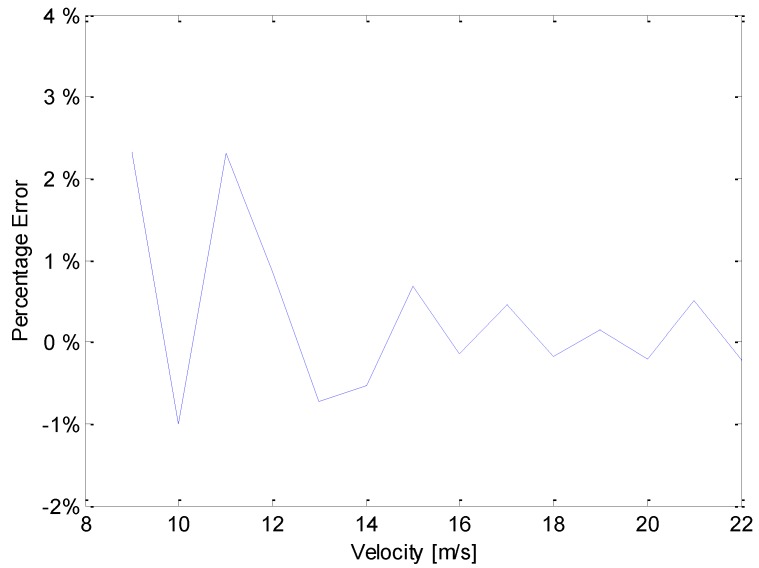
Percentage error analysis curve.

**Table 1 sensors-19-01657-t001:** Definition of boundary conditions.

Inlet	Velocity Inlet
Outlet	Pressure outlet
Pipe Surface	Symmetry
PVDF film and others	Non-slip wall

**Table 2 sensors-19-01657-t002:** Total sound level for different wind speeds.

Wind Speed (m/s)	10	15	20	25	30
Total sound level (dB)	57.66128	66.90901	74.39316	78.6987	83.82377

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
