# Peer review of "A Flow Velocity Measurement Method Based on a PVDF Piezoelectric Sensor"

_sensors, 2019, doi:10.3390/s19071657_

Round 1
Reviewer 1 Report
The manuscript "A Flow Velocity Measurement Method Based on a PVDF Piezoelectric Sensor” demonstrated the flow velocity sensor based on PVDF piezoelectric mechanical sensor array. The authors investigated relationship between flow velocity and the electrical output of the PVDF piezoelectric film, and the proposed method is shown to be reliable and effective. This is a good development in the field of piezoelectric sensor and flow velocity sensor. It has sufficient novelty and significance for the publication in Sensors. However, there also are several issues that the authors need to address carefully first
1. What is the equipment to measure electric signal from PVDF sensor?
2. What is the internal resistance of the measurement equipment ?
3. Why the PVDF produced negative voltage?
4. What is the thickness of the PVDF?
5. How is the quality of crystallinity of the PVDF prepared?
Author Response
Thanks for your questions.
Point 1: What is the equipment to measure electric signal from PVDF sensor?
Response 1: B&K PULSE Noise and Vibration Multi-analyzer System is used to do the measurement.
Point 2: What is the internal resistance of the measurement equipment ?
Response 2: 50MΩ
Point 3: Why the PVDF produced negative voltage?
Response 3: The voltage itself is positive, but because the value is very small, it is negative after evaluating the logarithm. This paper uses the voltage (in logs).
Point 4: What is the thickness of the PVDF?
Response 4: 1.5mm
Point 5: How is the quality of crystallinity of the PVDF prepared?
Response 5: 30%-50%, it meets the requirements of piezoelectric properties of PVDF films.
Reviewer 2 Report
The paper is well-written and presents an application with practical importance. More thorough literature review is required, since the usage of this type of sensors is relatively old in literature, although the reviewer could not find a similar application in open literature.
see for example: Wang, YC., Huang, CH., Lee, YC. et al. Exp Fluids (2006) 41: 365. https://doi.org/10.1007/s00348-006-0135-8
Author Response
Point 1: The paper is well-written and presents an application with practical importance. More thorough literature review is required, since the usage of this type of sensors is relatively old in literature, although the reviewer could not find a similar application in open literature.
see for example: Wang, YC., Huang, CH., Lee, YC. et al. Exp Fluids (2006) 41: 365. https://doi.org/10.1007/s00348-006-0135-8
Response 1: Thank you for your comments.Now more thorough literature review has been added.
Reviewer 3 Report
The paper quotes only Chinese references as if nonbody else developed PVDF sensors. The literature survey in the introduction is inadequate and goes too much into the history of PVDF and a bit trivial when hydrodynamic sensing is discussed. Section 1 should be thoroughly revised.
Equation 4 is correct if the strain is evenly distributed which is usually not the case in sensors. In addition, the letter S is used to designate the area. In piezoelectric notation S designates strain. The authors should change it to A designating Area.
Line 128 – Please pay attention to font size.
- Units are designated with a square parenthesis [N] not (N)
Fig. 4 – Please revise Fig. 4 for scientific publication.
Fig. 12 – What is the unit of the error
Section 4.2 – The experimental setup is not described properly.
All and all the paper do not describe a novel sensing system. In the 90-s already PVDF sensors have been used for flow measurement (which is not described in the introduction). The scientific novelty is very limited. The technical quality of the paper is not thorough enough. Many details are missing.
Author Response
please find the responses in the attachment.

Reviewer 4 Report
The authors demonstrated a PVDF piezoelectric film with patterned electrodes to design an array of sensors. The work presented is very preliminary and lacks novelty. In addition, the technical aspects of the sensor are not well explained.
Some major questions:
- How can a piezoelectric sensor be utilized here? In addition to the sound pressure, the sensor will experience a drag force due to the fluid flow over such a large surface area. How does the sensor behave in response to the drag force generated by the air flow. What are the effects of vortex-induced vibrations generated in the film and how can they influence the flow measurements?
- Most of the theory explained in the section 2 and simulations conducted in the manuscript are very well-known and preliminary in nature.
- The electrode design to collect voltage charges generated from the film is unclear. Since all the electrodes are patterned on the single PVDF film, it is unclear how the cross actuation and voltage generation between sensors could be distinguished from the actual individual sensor output.
- Although an array of sensors were developed no flow profile results were provided.
Author Response

(The authors gave the same response as above.)

Round 2
Reviewer 3 Report
The introduction is still not comprehensive enough. PVDF sensors were already used in MAVs: Yang et al. 2012.
The experiment in section 4.2 needs additional work. Please add the drawing where the sensor is placed and a schematic drawing f the setup would be useful too.
The paper needs further corrections
Author Response
Point 1: The introduction is still not comprehensive enough. PVDF sensors were already used in MAVs: Yang et al. 2012.
Response 1: According to your description, the paper titled Flapping wings with PVDF sensors to modify the aerodynamic forces of a micro aerial vehicle was found and it has been added to the introduction as ref.13 now.
Point 2: The experiment in section 4.2 needs additional work. Please add the drawing where the sensor is placed and a schematic drawing of the setup would be useful too.
Response 2: The drawing where the sensor is placed and a schematic drawing of the setup (Figs. 10 and 11) has been added to section 4.2 now.